

# Habitat composition near linear landscape structures across Poland: perspectives on pollinator conservation

Emilia Marjańska and Dawid Moroń

Institute of Systematics and Evolution of Animals Polish Academy of Sciences, Kraków, Poland

## ABSTRACT

Landscape management intensification is a major driver of global decline in insect pollinators and the ecosystem services they provide. Part of the proposed solution is to take advantage of the potential benefits of existing human-made habitats such as linear landscape structures (LLSs) associated with highways verges, railway embankments, or levees. We explored the surrounding landscape composition of different LLSs across Poland using geographic information system (GIS) methodology, examining spatial scales reflecting typical foraging distances of key pollinator groups (bees, butterflies, flies). We found that land cover composition around LLSs closely mirrored the overall national land cover distribution in Poland with right-skewed distributions of tree, grass, crop, and built cover across all LLS types. Highway verges exhibited the highest landscape diversity, while railways and levees showed more similar patterns to each other. Our study revealed that land cover changes occur unidirectionally at the studied scales, suggesting consistent habitat transitions around these linear features. Based on these findings, we propose prioritising two approaches for pollinator conservation in Poland: (1) using levee verges as supplementary habitat in wetland areas due to their lowest traffic level and reduced cost; (2) developing highway verges as a supplementary habitat to support pollinator ecosystem services in farmland and urban landscapes. Since LLSs are managed by relatively few stakeholders, primarily public bodies and government agencies, which provides an opportunity to implement management plans supporting biodiversity and ecosystem services at the national and regional scale.

## INTRODUCTION

Insect pollinators (bees, butterflies, and flies; hereafter, pollinators) provide essential ecosystem services (*Potts et al., 2010*). Landscape management intensification is a major driver of the global decrease in these pollinators and, consequently, pollination services (*Potts et al., 2016*; *Sánchez-Bayo & Wyckhuys, 2019*). Intensively managed landscapes are characterised by semi-natural habitat patches separated from each other by a matrix of agricultural fields or urban areas (*Hanski, 1989*; *Öckinger et al., 2012*). Metapopulation theory predicts and empirical studies confirm that both habitat loss and fragmentation contribute to local population extinctions (*Hanski, 1998*; *Baguette et al., 2013*). Thus, conservation strategies that counteract habitat loss and isolation should take into account

Corresponding author
Dawid Moroń,
dawidmoron@protonmail.com

landscape composition and spatial configuration, which are important determinants of biodiversity and ecosystem services (*Potts et al., 2016*; *Fattorini et al., 2018*).

Improving landscape suitability for pollinators and sustaining their vital ecosystem services present several practical problems. For example, habitat restoration depends on available land (*Strassburg et al., 2020*), landscape management is associated with funding (*O'Sullivan et al., 2017*; *Roberts & Phillips, 2019*), and the effectiveness of landscape interventions might depend on the traits of pollinator species being targeted and the type of matrix or landscape structure (*Li et al., 2020*; *Van Schalkwyk et al., 2022*). Part of the proposed solution is to utilise the unrecognised benefits of already existing habitats associated with artificial structures that alter the spatial composition and permeability of the landscape (*Moroń et al., 2014*; *Moroń et al., 2017a*; *Phillips et al., 2020b*). Highways, railways, and levees—linear landscape elements (LLSs)—offer such opportunities because of land associated with the LLS networks called verges (also known as embankments, nature strips, sidewalk buffer, parking strips, or tree lawn; *O'Sullivan et al., 2017*).

LLSs may have high potential for preserving plant and pollinator populations because they cover large areas as a habitat (*Villemey et al., 2018*; *Phillips et al., 2021b*). Moreover, the specific structure of most LLSs, *i.e.,* a steep embankment with a dry, insolated area at the top, and a wetter area at the bottom, creates a strong environmental gradient that may favour different flowering plants that attract diverse pollinator communities (*Moroń et al., 2014*; *Moroń et al., 2017a*; *Bátori et al., 2016*; *Heneberg, Bogusch & Řezáč, 2017*; *Monasterolo et al., 2020*; *Twerd, Sobieraj-Betlińska & Szefer, 2021*), thereby increasing overall ecosystem services (*Phillips et al., 2020a*). Additionally, the linear shape of the LLSs makes associated habitats continuous and intersects the landscape or even regions, potentially increasing habitat connectivity, as opposed to patchy systems of semi-natural habitat remnants (*Plue et al., 2022*). Importantly, the LLSs are managed by relatively few stakeholders, very often by public bodies and government agencies, which provides an opportunity to implement management plans supporting biodiversity and ecosystem services at the national or regional scale. As a relatively mobile species, pollinators require resources at a landscape scale that extends beyond LLS verges. This necessitates a comprehensive assessment framework to evaluate landscape characteristics important for pollinator across different LLS types. However, to our knowledge, there has been no formal quantitative assessment of landscape that surrounds the different LLSs at a country level in the European Union (EU) (for roads see: *Phillips et al., 2021a*; *Phillips et al., 2021b*). Such a comparison between different LLSs is important to identify which LLS should be prioritised for nature-based landscape management. It can also provide practical insights into how LLSs can be used for increasing pollination services as a supplementary habitat based on their landscape characteristics (*Mallinger, Gibbs & Gratton, 2016*; *Hyjazie & Forrest, 2024*). For example, LLSs in human-modified landscapes may boost pollinator populations by providing necessary food or nesting resources, which are scare or even absent in the landscape, whereas in semi-natural landscapes, LLSs boost pollinator populations by increasing the availability of resources over time or their diversity (*Dunning, Danielson & Pulliam, 1992*). Management plans such as increase food for key pollinators, or pollinator biodiversity such

as increase habitat diversity at landscape scales, in such supplementary habitats might be directed toward pollinator services (*Müller et al., 2024*).

In the EU, there are more than 200,000 km of railway tracks (*Statista Search Department, 2023a*), 75,000 km of highways (*Statista Search Department, 2023b*) and at least 38,000 km of levees (data for Czechia, France, Germany, Hungary, Netherlands and Poland; *Tournament et al., 2018*), which demonstrates that LLSs are common features of the landscape. Grass and tree covers (providing food and nesting sites for most species) are landscape types positively affecting pollinators, whereas, crop and built covers might negatively be associated with pollinators (filtering pollinator species adapted to human-modified environments; *Potts et al., 2015*). In addition, more complex landscapes, *i.e.,* containing different types of land covers, support more diverse pollinator communities (*Bottero et al., 2023*). Thus, a potential role of LLSs as supplementary habitats to support pollinator populations and its biodiversity-related ecosystem services results from landscape characteristics. Therefore, we carried out a comprehensive assessment of the landscape surrounding the LLSs across all of Poland as a first step towards determining the potential of LLS verges as supplementary habitats to enhance pollinator populations (*Sánchez-Clavijo, Bayly & Quintana-Ascencio, 2020*). We used stratified random sampling of LLS vector maps (*Geoportal, 2023a*) with a combination of land cover maps of high resolution (*Zanaga et al., 2022*). Specifically, we explored (1) the surrounding landscape of LLSs in comparison to Poland's landscape, (2) the surrounding landscape composition of LLSs according to spatial scales (buffer of 250, 500, and 1,000 m around LLS), (3) the surrounding landscape of the LLSs according to cover distribution shapes, and (4) the surrounding landscape diversity of LLSs. From this information, we identified potential opportunities be prioritised with respect to a particular LLS as a supplementary habitat for the conservation and management of pollinator diversity according to landscape type, *e.g.*, semi-natural *vs.* human-modified, or according to demand for ecosystem services, *e.g.*, crops *vs.* non-crops. The findings of this study are important for preliminary identification of the current and future capacity of different LLSs to benefit nature and the environment at the national and regional levels. We provide practical insights for selecting different LLSs as multifunctional green spaces, with particular relevance for areas highly modified by human interventions where land scarcity is a serious limitation for nature conservation and ecosystem services.

## MATERIALS AND METHODS

### Databases and definitions of the LLSs

To identify the surrounding landscape composition of LLSs in Poland (Fig. 1A), we used freely available databases from Geoportal (*Geoportal, 2023a*) and the European Space Agency (ESA; *European Space Agency, 2023*). Geoportal provides a database (Topographic Object Database: BDOT10k) containing the spatial location of topographic features in Poland (Figs. 2A–2C) along with descriptions of their properties (*Geoportal, 2023b*). ESA provides a database (WorldCover V2 2021) with global land cover products for 2021 at 10 m resolution, which were developed and validated in near-real time based on Sentinel-1 and Sentinel-2 satellite data (*Zanaga et al., 2022*). Land cover was independently validated with

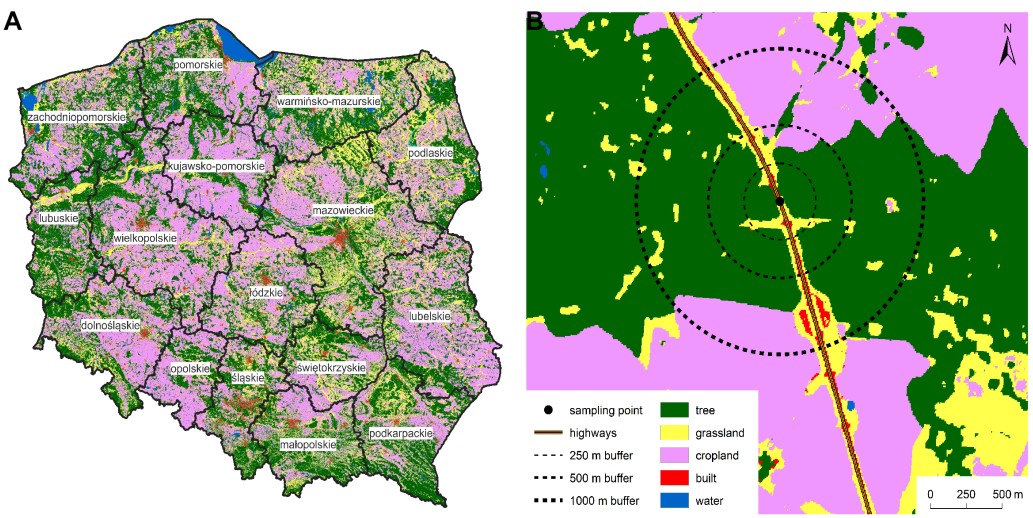

**Figure 1** **Map of Poland divided into 16 voivodeships with a sample point at the linear landscape structure and designated buffers.** Map of Poland divided into 16 voivodeships (names labelled) with land cover according to the WorldCover V2 2021 database (A). Map of a sample point at the linear landscape structure (highway) and designated buffers with the land cover (B). © ESA WorldCover project 2021/contains modified Copernicus Sentinel data (2021) processed by ESA WorldCover consortium.

a global overall accuracy of 76.7% (*Zanaga et al., 2022*). The WorldCover database, which covers the whole world, enables applying the approach presented in the study to other countries and regions with a possibility to compare results. Thus, we obtained Geoportal-based datasets for railway, highway, and levee identification. We divided Poland into 16 administrative regions (voivodeships) and used ESA-based datasets for the assessment of land cover surrounding LLSs.

We used the objects of layers SKDR01 (highways) and SKDR02 (express roads; Fig. 2A) for the study based on the BDOT10K database. The railways were defined as objects of layer SKTR0, excluding railway sidings (Fig. 2B). Levees were objects of the layer BUZM02 (Fig. 2C).

## LLS sampling

We used stratified random sampling to assess the variation in landscape composition of the LLSs across Poland. A total of 7,500 random sampling points were generate, stratified across the 16 strata (16 regions of Poland: voivodeships; Figs. 2D–2F). The number of points per stratum was determined by the length of an LLS in a stratum as a proportion of the total length of the LLSs for Poland. The median point number per stratum was 32 for highways (range: 11–73; total: 573; Fig. 2D), 201 for railways (125–483; 3,821; Fig. 2E), and 163.5 for levees (48–424; 3,106; Fig. 2F). The minimum distance between sampling points within each stratum was two km (distance to nearest point (median); railways: 2,873 m; levees 3,259 m; highways: 4,194 m).

For each sampling point, we extracted the coordinates (longitude and latitude), LLS class (railway, highway and levee), administrative region (16 voivodeships), and landscape characteristics.

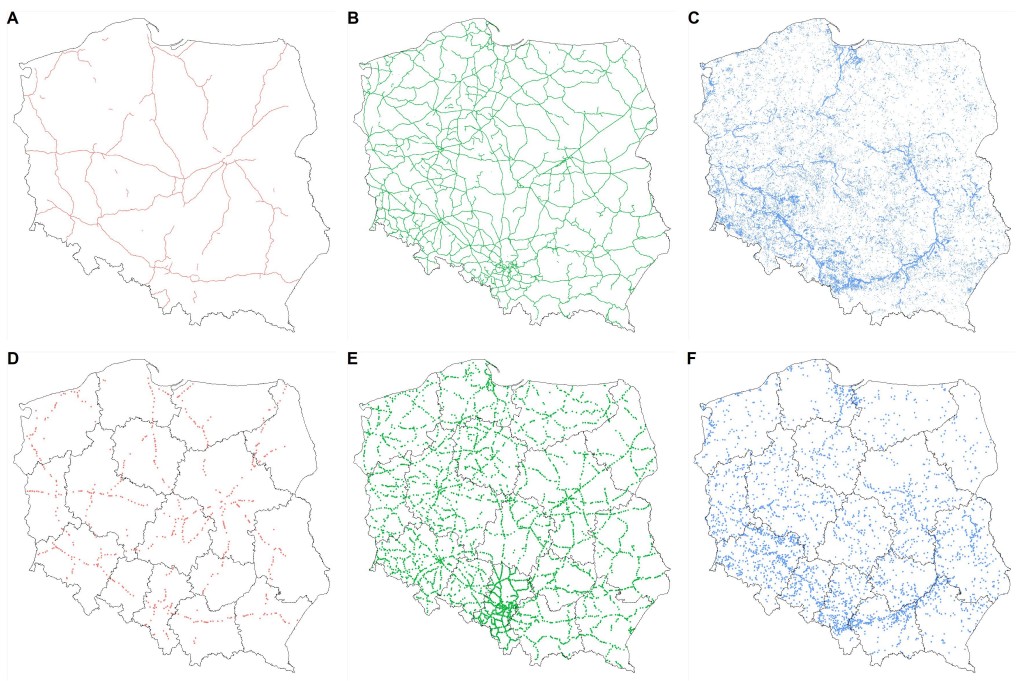

**Figure 2** **Maps of linear landscape structures in Poland.** Maps of linear landscape structures in Poland: highways (A), railways (B), and levees (C). Maps of the sampling points at highways (D; 573 sampling points), railways (E; 3,821), and levees (F; 3,106); overall, 7,500 sampling points. Administrative regions of Poland (16 voivodeships; (D–E)) within each linear landscape structure formed 16 strata for stratified sampling. The number of sampling points per stratum was determined by the length of a linear landscape structure in the stratum as a proportion of the total length of a linear landscape structure for Poland.

## Measuring landscape characteristics of the LLSs

We summarised the surrounding land use by creating three buffers (250 m, 500 m, 1,000 m) around the overall LLS lines of each type as well as around each sampling point (Fig. 1B) and then extracting the percentage of a land use class within the buffers using WorldCover V2 2021. Selected buffers correspond to distances covered by most pollinators (*Zurbuchen et al., 2010*), especially bees, which are central place foragers (*Michener, 2000*). Moreover, landscape characteristics affect most pollinators at referred scales of 250–1,000 m (*Steffan-Dewenter et al., 2002*; *Saturni, Jaffé & Metzger, 2016*; *Lajos et al., 2021*; *Misiewicz, Mikołajczyk & Bednarska, 2023*; *Kammerer et al., 2024*). Distance covered by pollinators is another important life history trait affecting pollination efficiency (*Abrol, 2012*). We treated the 250 m buffer as the reference scale because many small-bodied species cover only short distances, *i.e.,* of less than 300 m (*Gathmann & Tscharntke, 2002*; *Zurbuchen et al., 2010*).

We used all 11 land use classes of WorldCover V2 2021 to give an overall percentage area within each buffer that consisted of land use categories: "bare/sparse vegetation", "built-up", "cropland", "grassland", "herbaceous wetland", "mangroves", "moss and lichen", "permanent water bodies", "shrubland", "snow and ice", and "tree cover" (*Zanaga et al., 2022*). However, at the scale of Poland, there is no "mangrove" class, and the cover for some

classes is lower than 5 km$^2$ ("moss and lichens"–4.85 km$^2$, "shrubland"–0.25 km$^2$, "snow and ice"–0.06 km$^2$). Thus, these classes were excluded from the analysis and visualisation of the data. The most predominant land use classes were tree cover (127.0 * 10$^3$ km), crop cover (109.8 * 10$^3$ km), grass cover (60.1 * 10$^3$ km), and built cover (8.7 * 10$^3$ km).

### Data analysis and visualisation

The inverse Simpson (D) index of landscape cover diversity was calculated for each sampling point to quantify landscape diversity. The value of this index starts with one representing a landscape containing only one land cover type. The higher the value is, the greater the diversity. The maximum value is the number of land cover types; here seven land cover types in this study.

For assessing differences in landscape covers between LLSs we estimated effect sizes using Hedge's g or Cohen's d formulas. The Hedge's g formula with bias correction was used when the sample size was less than 30, *i.e.,* for surrounding landscape of LLSs *vs.* Poland's landscape; otherwise, we used Cohen's d formula (*Khan & McLean, 2024*). The paired Cohen's d was used for analysing landscape of LLSs according to three spatial scales; otherwise, we used unpaired formulas. Effect sizes were estimated with 95% confidence intervals (CI).

Differences in distributions of standardised covers between LLSs were investigated with Kolmogorov–Smirnov test. By, visualising differences between empirical distribution functions (ECDF) of LLSs landscape covers, we could identify at what cover values the differences were the greatest. The shape of distributions may be described with help of skewness, which is a measure of distribution asymmetry. A right-skewed distribution shape indicates that most of the surrounding landscape of sampling points are characterised by relatively small cover values, *i.e.,* less than mean/median. In contrast, a left skewed distribution shape indicates that most of surrounding landscape of sampling points are characterised by cover values larger than mean/median.

The database management and map visualisation were performed using ArcGIS 10.8.2 (*Environmental System Research Institute, 2020*) and QGIS 3.22 (*QGIS Development Team, 2022*).

All data analysis and visualizations were undertaken using R ver. 4.3.0 (*R Core Team, 2023*) and the following packages: Durga (*Khan & McLean, 2024*), dyplr (*Wickham et al., 2023*), ggplot2 (*Wickham, 2016*), ggpubr (*Kassambara, 2023*), jpeg (*Urbanek, 2022*), moments (*Komsta & Novomestky, 2022*), readr (*Wickham, Hester & Bryan, 2024*), and vegan (*Oksanen et al., 2022*).

## RESULTS

### Surrounding landscape of LLSs *vs.* Poland's landscape

Land cover within buffers of 250 m around highways, railways, and levees corresponded to the overall proportion of land cover in Poland (Fig. 3). The landscapes most different from the median land cover for Poland were tree cover within highway buffer (0.41 *vs.* 0.25, Poland *vs.* LLSs; Fig. 3A), grass cover within levee buffer (0.18 *vs.* 0.28; Fig. 3B), crop cover within levee buffer (0.30 *vs.* 0.22; Fig. 3C), built cover within highway buffer (0.03 *vs.*
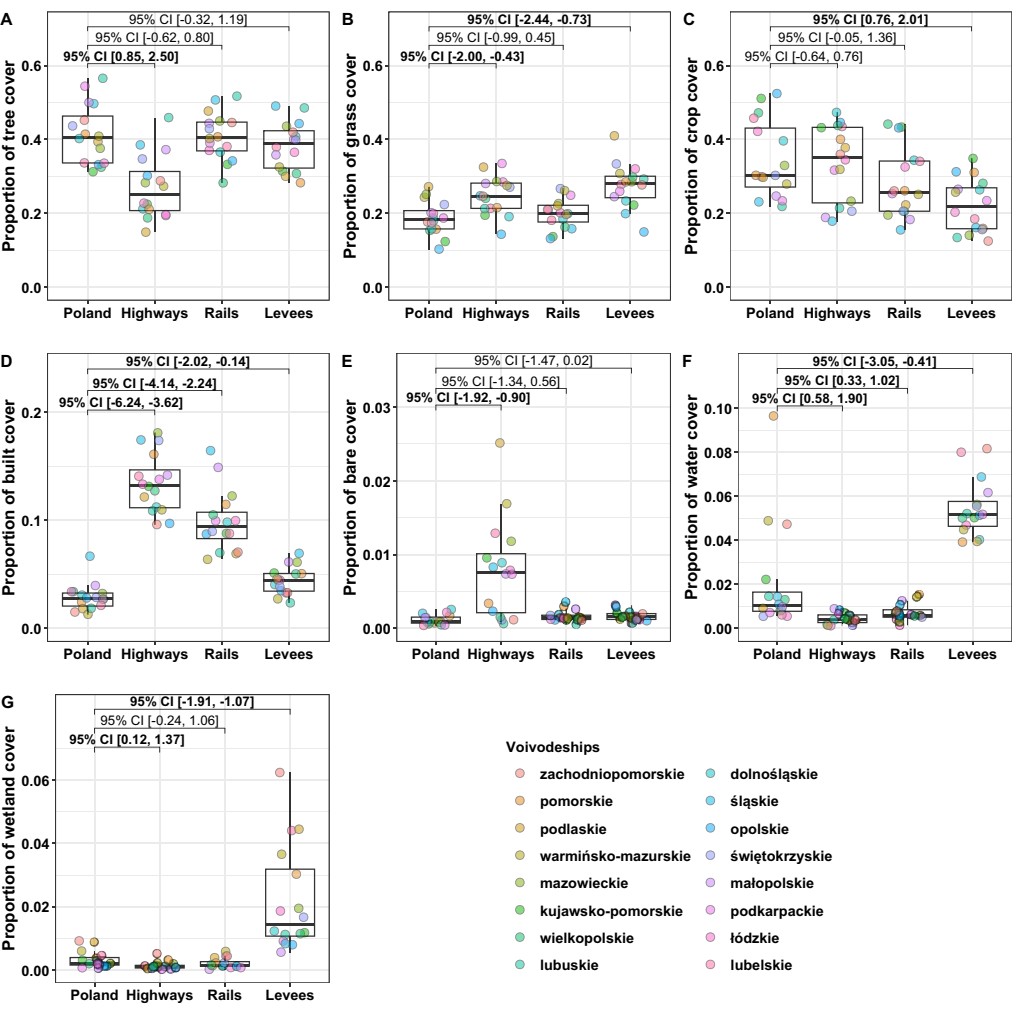

**Figure 3  Cover of tree, grass, crop, built, bare, water, and wetland within buffers of 250 m around linear landscape structures.** Cover of tree (A), grass (B), crop (C), built (D), bare (E), water (F), and wetland (G) within buffers of 250 m around linear landscape structures (highways, railways, and levees). The land cover of Poland was measured excluding linear landscape structures and buffers. Measures of land cover were taken separately for each of the voivodeships of Poland ($N = 16$; highlighted by point colours). The top and bottom boundaries of each box indicate the 75th and 25th quartile values, respectively. The black lines within each box represent the median values. Values at brackets indicate 95% confidence intervals (CI) of effect sizes (unpaired Hedge's g); bold values are CIs not overlapping zero.

0.13; Fig. 3D), bare cover within highway buffer (0.0009 *vs.* 0.0076; Fig. 3E), water cover within levee buffer (0.01 *vs.* 0.05; Fig. 3F), and wetland cover within levee buffer (0.002 *vs.* 0.014; Fig. 3G).

## Surrounding landscape of LLSs according to spatial scales

The cover of the most common land types (cover of tree, grass, crop and built) in selected buffers (250 m, 500 m, and 1,000 m) around LLS sampling points changed mostly in a unidirectional manner. The land cover of trees (0.147, 0.212, and 0.267 for 250 m, 500 m,

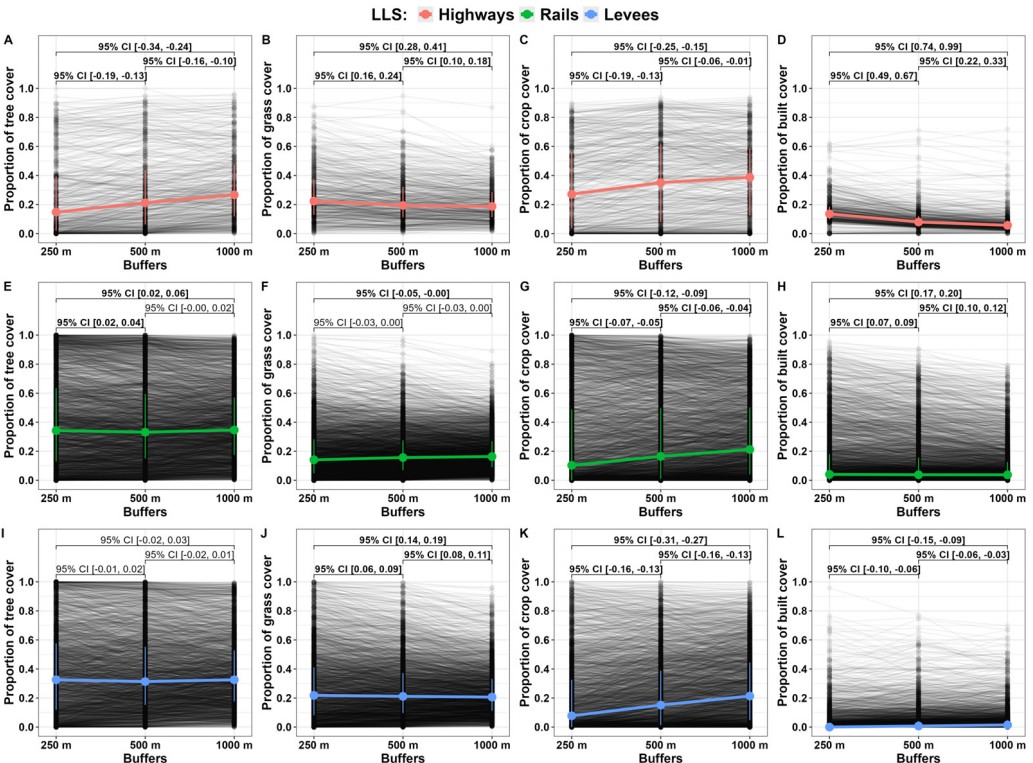

**Figure 4  Changes in tree, grass, crop, and built cover according to buffer size (250 m, 500 m and 1,000 m) around randomly selected points at linear landscape structures.** Changes in tree (A, E, I), grass (B, F, J), crop (C, G, K), and built (D, H, L) cover according to buffer size (250 m, 500 m, and 1,000 m) around randomly selected points ($N = 7,500$) at linear landscape structures (highways, rails, and levees). Highlighted points indicate median values and whiskers indicate the 75th and 25th quartile values. Values at brackets indicate 95% confidence intervals (CI) of effect sizes (paired Cohen's d); bold values are CIs not overlapping zero.

and 1,000 m buffers, respectively), and crops (0.271, 0.351, and 0.387) around highway buffers increased with the buffer size (Figs. 4A, 4C). In contrast, grass (0.224, 0.194, and 0.187) and built cover (0.135, 0.080, and 0.057) decreased (Figs. 4B, 4D). The rail buffer cover of tree (0.342, 0.331, 0.347) and grass (0.141, 0.157, 0.163) were not affected by buffer size (Figs. 4E, 4F); however, crop (0.103, 0.165, 0.212) cover increased (Fig. 4G) and built (0.041, 0.038, 0,038) cover decreased with buffer size (Fig. 4H). The levee buffer area did not affect tree (0.326, 0.313, 0.326) cover (Fig. 4I); however, the cover of crops (0.078, 0.151, 0.214) and build (0.000, 0.007, 0.015) increased (Figs. 4K, 4L) but grass (0.218, 0.212, 0.206) decreased with buffer size (Fig. 4J). Across LLS types, only crop cover uniformly changed, *i.e.,* increased with buffer size (Figs. 4C, 4G, 4K).

## Surrounding landscape of the LLSs according to cover distribution shapes

The distributions of tree, grass, crop, and built cover were highly right-skewed for all LLSs 250 m buffers around the sampling points (Fig. 5). This distribution shape type indicates that most of the surrounding landscape of sampling points have relatively small cover

values, *i.e.,* less than mean. The tree cover distribution shape was similar for rail (skewness = 0.483) and levee (0.572) buffers; however, for highways (0.971), the share of tree cover decreases more steeply (Fig. 5A). The distribution shapes of grass cover within 250 m buffers of highway (0.900), rail (1.333), and levee (0.976) sampling points were similar to each other (Fig. 5B). Considering the distribution of crop cover around the 250 m sampling point buffers, the distributions of rails (0.955) and levees (1.326) are similar, but highways (0.356) were characterised by a larger share of 0.25–0.75 crop cover (Fig. 5C). Although the distribution shape of built cover within 250 m buffers rapidly decreased for levees (4.166), the declines for highways (1.578) and railways (1.839) were lower (Fig. 5D). Differences between ECDF of LLS landscape covers were the greatest at the low cover values (below 50%; Supplement S1), except tree cover for rails and levees where the ECDF differences were the greatest at the high tree cover (above 50%; Supplement S1C).

### Surrounding landscape diversity of LLSs

The inverse Simpson index revealed that the landscape diversity of seven land covers (Fig. 3) within a 250 m buffer around sampling points was the highest for highways ($D = 2.44$; Fig. 6), low for levees ($D = 2.23$; Fig. 6), and the lowest for railways ($D = 2.03$; Fig. 6).

# DISCUSSION

There is a growing need for using novel habitats, such as LLS verges, for nature conservation and to promote ecosystem service provision (*Moroń et al., 2017a*; *Skórka, Lenda & Moroń, 2018*; *Kajzer-Bonk et al., 2019*; *Phillips et al., 2021b*), especially in highly artificially modified landscapes. This study is a first step towards analysing landscape composition surrounding LLSs to assess the potential of highways, railways, and levees to support pollinator populations as supplementary habitats at a national scale. Here, we discuss the potential of different LLSs, accounting for their surrounding landscape, to alleviate conflict between biodiversity and civilisation demands.

### Surrounding landscape of LLSs *vs.* Poland's landscape

Landscape composition surrounding LLSs well mirrored land use types at the national scale (Fig. 3). Most pollinator hotspots in temperate climates are linked with open habitats, *i.e.,* grass cover. Grasslands are under severe pressure, as traditional land use is increasingly replaced by either highly intensive agricultural practices or abandonment (*Noordijk et al., 2009*). Thus, mostly grassy levee and highway verges can potentially support diverse communities of pollinators as a supplementary habitat due to the larger proportion of grass cover in the surroundings compared to Poland's cover (Fig. 3B). However, woodlands can also act as a habitat for pollinator communities (*Alison et al., 2022*); therefore, mostly grassy rail verges can be considered as a supplementary habitat likely boosting pollinator diversity due to the larger proportion of wood cover in the surroundings compared to Poland's cover (Fig. 3A).

Wetlands are threatened habitats worldwide due to changes in land use, especially in the EU and North America (*Davidson, 2014*). The highest proportion of the habitat consisted of buffers surrounding levees (Fig. 3G). Thus, actions considering wetland conservation and

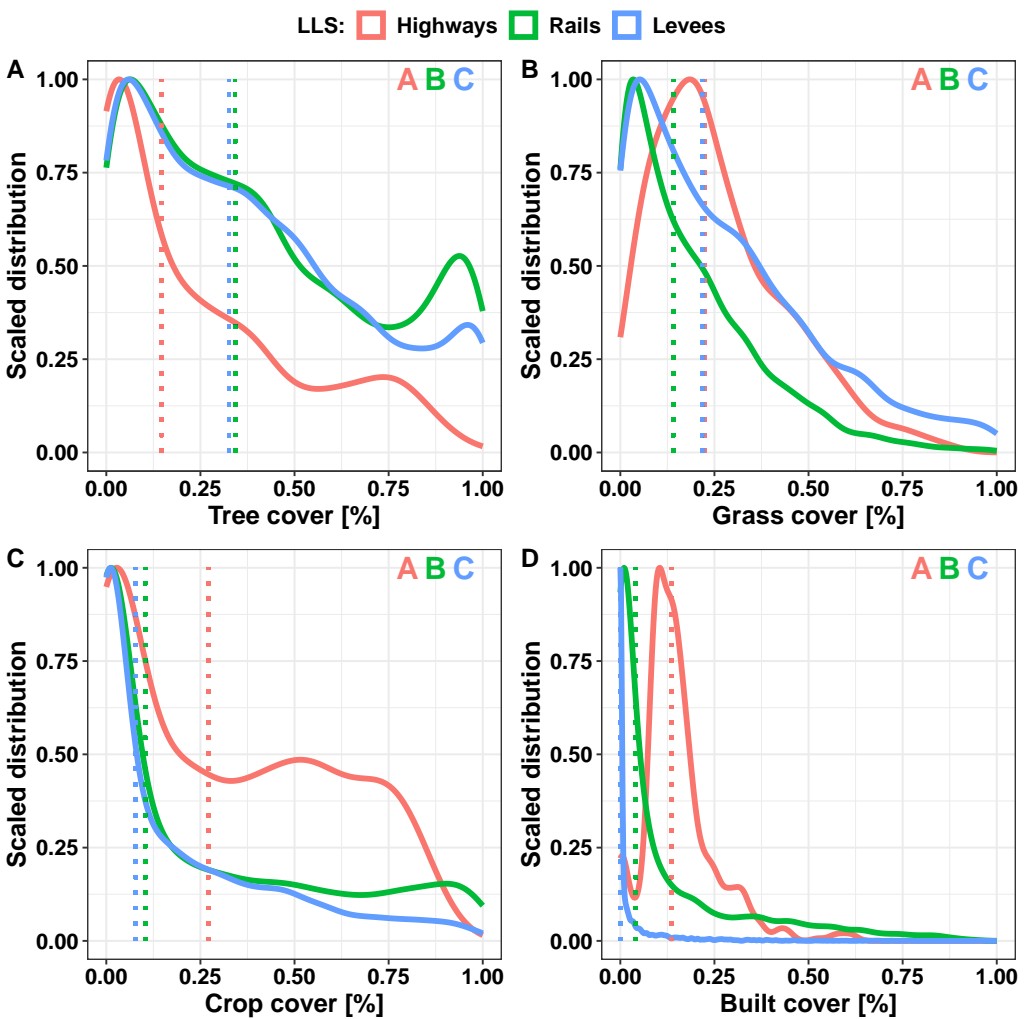

**Figure 5** **Scaled distribution of tree, grass, crop, and built cover within buffers of 250 m around randomly selected points at linear landscape structures.** Scaled distribution of tree (A), grass (B), crop (C), and built (D) cover within buffers of 250 m around randomly selected points ($N = 7,500$) at linear landscape structures (highways, rails and levees). Dotted lines indicate the median value of a cover. Different letters indicate significance ($p < 0.005$), letter colour correspond to an linear landscape structures type.

restoration should include dryer and unflooded levee verges as a supplementary habitat. Interestingly, although most bee species are ground nesters, wetland areas may be rich in many bee species (*Moroń et al., 2008*).

From food safety and economy perspectives, pollinator services are mostly linked to commercial crops that are dependent on pollination (*Potts et al., 2010*; *Abrol, 2012*). Among the three studied LLSs, highway verges may provide pollination services to the largest proportion of crop land cover (Fig. 3C). Thus, landscape managers should identify the potential of highway verges as a supplementary habitat to increase pollinator services provided to nearby crops. Additionally, highways are surrounded by landscapes with a higher proportion of built cover than the other LLSs. In light of climate change and

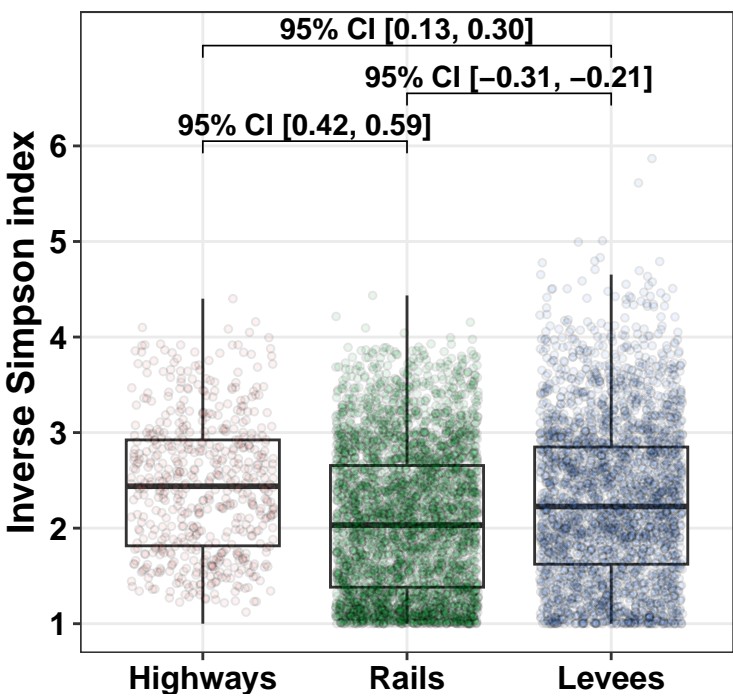

**Figure 6** **Landscape diversity measured by inverse Simpson within buffers of 250 m around randomly selected points at linear landscape structures.** Landscape diversity measured by inverse Simpson within buffers of 250 m around randomly selected points ($N = 7,500$) at linear landscape structures (highways, railways, and levees). The top and bottom boundaries of each box indicate the 75th and 25th quartile values, respectively. The black lines within each box represent the median values. Values at brackets indicate 95% confidence intervals (CI) of effect sizes (unpaired Cohen's d).

predicted food security concerns (*Van der Sluijs & Vaage, 2018*), growing food in and around cities could also be a partial solution to sustainably increasing food production in an urbanised world (*Nicholls et al., 2023*). Thus, the potential of highway verges should be considered as a potential supplementary habitat in human-dominated landscapes.

## Surrounding landscape of LLSs according to spatial scales

Bees, the most important pollinators in the temperate zone (*Abrol, 2012*), as central placed foragers, use only part of the available landscape because they cover limited distance (*Zurbuchen et al., 2010*). Distances realised by foraging bees are a function of the species body size, which translates into longer foraging routes of large-bodied bees (*Greenleaf et al., 2007*). Typically, most bee species cover up to 500 m from the nest; however, some social species, *i.e., Apis* spp. or *Bombus* spp., are able to cover more than one km (*Steffan-Dewenter et al., 2002*).

Flies and butterflies are not central place foragers and are potentially able to cover longer distances to search for a suitable patch. However, these pollinators avoid crossing a patch boundary (*Moroń, Skórka & Lenda, 2019*) and leave a patch only once the cost of continued foraging outweighs the amount of energy gained (*Tillman & Adelman, 2023*). This finding translates to the high range of distances covered by different butterfly species (*Skórka et al.,*
_2013b_; _Nowicki et al., 2014_; _Stefanescu et al., 2016_); for flies, no such systematic studies are known.

Many studies indicate that environmental factors affecting pollinator populations operate locally, _i.e.,_ floral diversity (_Moroń et al., 2014_), or concern the closest landscape, _i.e.,_ distance to the nearest semi-natural habitat (_Skórka et al., 2013a_), rather than at a broader landscape scale. Our study shows that the land cover surrounding LLSs is relatively uniform at the 250 m, 500 m, and 1,000 m scales (Fig. 4). This means that conservation programs at LLSs can target pollinators with different life history traits connected to habitat preferences, (_e.g.,_ nesting in wood) and mobility (_e.g.,_ body size). Additionally, geographic information system (GIS)-based plans that consider the suitability of LLSs for pollinators as supplementary habitats may be limited to one spatial scale, _e.g.,_ 250 m, saving time and reducing the analysis cost (_Rands & Whitney, 2011_).

## Surrounding landscape of LLSs according to cover distribution shapes

The distribution of land cover is highly right-skewed (Fig. 5), and there are many sampling points whose surrounding landscape is characterised by relatively small cover values, _i.e.,_ less than mean. Moreover, the shapes of the distributions are more similar between railways and levees than for highways (Fig. 5). Taking into consideration the shape of the land cover distributions, railways and levees seem to be equivalent, whereas highways have a more distinct distribution of land cover proportions. Thus, plans targeting pollinator diversity and their ecosystem services in LSSs as supplementary habitats should consider the potential differences in landscape cover distribution patterns and the feasibility of using different types of LLSs as alternatives, given that the three studied LLSs share a structure. The relatively larger proportion of LLSs with smaller values of the covers (_i.e.,_ right-skewed distributions) may imply that most LLS management schemes may affect several land cover types.

## Surrounding landscape diversity of LLSs

An important characteristic of landscapes that support biodiversity is habitat complexity (_Bottero et al., 2023_). If a landscape is more diverse, more potential food resources and nesting sites are available for pollinators (_Gathmann & Tscharntke, 2002_). Moreover, among pollinator species, many have narrower food or nesting preferences; thus, available habitats are even more needed for specialised species (_Hadley & Betts, 2012_). Here, we show that the probability of containing more land use types in the near surroundings is higher for highways than for other studied LLSs (Fig. 6).

Unfortunately, predicting the effects of additional habitat types in the landscape on pollinator richness and abundance is not trivial, mostly because ecological processes seem to be idiosyncratic, _i.e.,_ include nonlinear relationships (_Moroń et al., 2019_). However, knowledge about the more diverse land cover of the surrounding LLSs can be incorporated into ecological management programs. On the basis of our findings, we emphasise that highways should be prioritised as a supplementary habitat to sustain pollinator populations and their ecosystem services, considering the landscape diversity.

## CONCLUSIONS

Landscape composition surrounding LLSs relatively well mirrored land use types at the national scale; however, there were some exceptions. For example, the highest proportion of wetlands consisted of buffers surrounding levees or urban and farmland surrounding highways. Our study shows that the land cover surrounding LLSs changes mostly in a unidirectional manner at the 250 m, 500 m, and 1,000 m scales. Considering the shape of the land cover distributions, railways and levees seem to be relatively similar. However, we show that the probability of containing more land use types in the near surroundings is higher for highways than for other studied LLSs.

Accordingly, the preliminary prioritisation of LLSs as supplementary habitats in Poland for pollinators and their ecosystem service protection is as follows: (1) levee verges as having the lowest traffic level, thus lowering the potential cost of action, (2) levee verges for the pollinator conservation at wetlands (Fig. 4G), (3) highway verges for pollinator ecosystem services to crop in farmland (Fig. 4C), highway verges, followed by rails, for pollinator conservation at urban areas (Fig. 4D), (4) highway verges, followed by levees, for pollinator conservation for more complex landscapes (Fig. 5).

LLSs in general are not recognised as potentially important for shaping species diversity and ecosystem services at the landscape scale, in part due to data showing that some LLSs (*e.g.*, roads including highways) are sources of animal mortality and are a barrier when crossing (*Skórka et al., 2015*; *Remon et al., 2018*). Thus, their value as a habitat is questioned (*Murcia et al., 2014*; *Borda-de água et al., 2017*). Having recognised some positive aspects of LLSs for pollinators, *e.g.*, habitats and corridors (*Moroń et al., 2017a*; *Moroń et al., 2017b*), the possible threats to pollinator biodiversity should also be considered (*Phillips et al., 2020b*; *Phillips et al., 2021a*) when LLSs are recommended as potential elements of green networks. This indicates that the potential influence of LLSs is even greater and that the application of alternative, biodiversity-friendly methods of management may increase the positive role played by LLSs as supplementary habitats.

Estimating LLS's verge area is as rare as assessing LLSs' impact on the biodiversity of surrounding habitats (*Phillips et al., 2021b*). According to Polish regulations, there are at least six m from tracks on both sides of a rail line (*Sejm of the Republic of Poland, 2019*) and three m from embankment bases on both sides of a levee (*Sejm of the Republic of Poland, 2017*), which is restricted for land use, for example, planting trees (for highways, there is no direct regulation concerning the minimal distance of tree planting from the road edge, but highway verges are usually more than 50 m from both sides; GDDKiA's information provided). According to rough and conservative estimations, there is at least 1,057 km$^2$ of land surrounding LLSs in Poland subjected to some regulations of vegetation management, *i.e.,* tree planting. This amounts to 0.33% of the country's land. In contrast, Poland has almost 3,300 km$^2$ of national parks (*Statistics Poland (GUS), 2021*), which is approximately 1% of land cover, and the largest park covers 592 km$^2$. Therefore, even if the LLSs have relatively minor local contribution, the potential of using LLSs at the country scale seems to be substantial (*Riva & Fahrig, 2023*).

Marjańska and Moroń (2025), *PeerJ*, DOI 10.7717/peerj.19765

The studied LLSs considerably differ in disturbances caused by traffic (*Phillips et al., 2021a*). For highways, in Poland, there are up to 200,000 vehicles per day; for railways, there are usually fewer than 200 trains per day; and for levees, there is no traffic (excluding agricultural machinery commuting to farmlands and cyclists, some levees in Poland have paved surfaces for bicycle paths). As the cost of alleviating the negative effects of disturbance increases with disturbance level (*Trumbore, Brando & Hartmann, 2015*), among studied human-made LLSs, levees should be prioritised as supplementary habitats. However, prioritising other studied LLS for acting as supplementary habitats should also be considered as landscape surrounding LLSs can differ (Figs. 3–6).

Notably, the land cover distribution patterns for LLSs can differ between EU countries and their regions (*Phillips et al., 2021b*). It may be an effect of more or less complex land covers as well as the number of available habitats, governmental planning, or agricultural policy. However, Poland seems to represent a relatively good example of the EU's landscape because of the country size and history of land use common for Central Europe (*Lenda et al., 2018*). Typical landscape compositions of Poland are characterised by mosaics of tree, crop, and grassland cover (Fig. 1).

We made some general recommendations for pollinator populations in LLSs as supplementary habitats based on land covers. Relationships of pollinator populations and landscape characteristics are relatively well known and partially allow for such large-scale recommendations (*Sirohi, Jackson & Ollerton, 2025*). However, for comprehensive validation of the recommendations, we need more information about biodiversity of pollinators, their behaviour in LLSs, and GIS data covering more detailed land classifications. In particular, more studies are needed to assess how LLSs are changing spatial composition and configuration of the landscape considering habitat fragmentation and connectivity for animal movements (*McGarigal & Marks, 1995*). Future projects should consider landscape features (*e.g.*, permeability or food base distribution), pollinator species traits (*e.g.*, body size or food specialisation), systematic identity of species (*e.g.*, bees or flies), and their interactions to adjust spatial scales of landscape interventions to targeted pollinator species (*Lajos et al., 2021*). Thus, some more spatially restricted species (*e.g.*, small-bodied) or less spatially restricted species (*e.g.*, migratory) might be limited by landscape features at smaller or larger scales, respectively, than referred here (250–1,000 m). Importantly, we need more projects whose results can be scaled from a country level to a voivodeship or a municipality level where management schemes can be adapted to local conditions (*e.g.*, traffic level and landscape configuration). Moreover, greening of LLSs should be incorporated in future studies since it could be a good example of land sharing policy in urban and agricultural development, reducing human-nature conflict by combining active management and using the area for people's transportation, providing biodiversity and ecosystem services, and benefiting human wellbeing (*Moroń et al., 2024*).

### Funding
This research was financially supported by a grant from the National Science Centre, Poland (2020/37/B/NZ8/01743). The funders had no role in study design, data collection and analysis, decision to publish, or preparation of the manuscript.

### Grant Disclosures
The following grant information was disclosed by the authors:
National Science Centre, Poland: 2020/37/B/NZ8/01743.

### Competing Interests
The authors declare there are no competing interests.

### Author Contributions
- Emilia Marjańska conceived and designed the experiments, performed the experiments, analyzed the data, prepared figures and/or tables, authored or reviewed drafts of the article, and approved the final draft.
- Dawid Moroń conceived and designed the experiments, performed the experiments, analyzed the data, prepared figures and/or tables, authored or reviewed drafts of the article, and approved the final draft.

### Data Availability
  The raw measurements of land covers are available in the Supplementary File.

### Supplemental Information
Supplemental information for this article can be found online at http://dx.doi.org/10.7717/peerj.19765#supplemental-information.

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
