# Peer review of "Habitat composition near linear landscape structures across Poland: perspectives on pollinator conservation"

_PeerJ, doi:10.7717/peerj.19765_

## Round 0.1 · original submission · Major Revisions

· Academic Editor

Major Revisions

Dear authors, I kindly ask you to carefully correct and supplement the manuscript in accordance with the comments of the six reviewers. It is necessary to be careful in the wording, since you have not conducted a study of pollinators in the field. However, I believe that this article can be useful for organizing the protection of perennial pollinators. I hope that the second review of the corrected version of this manuscript will allow all reviewers to recommend it for publication.

·

Basic reporting

I really like this paper and think that it poses an original and interesting question.

This is an interesting manuscript that analyses habitat composition near linear landscape structures (levee verge, highway verges, railway) across Poland and recommendations for pollinator management. Based on their results, they identified opportunities to improve the management of linear landscape elements to provide pollinator diversity and ecosystem services.

I conclude that the manuscript fits the scope of the journal.
The structure of the article is to follow an acceptable journal format.
The abstract is clear and well written. Introduction provide sufficient background.The objective of the paper is clearly explained.
The data appear appropriately collected and analysed. The analysis is mostly sound and the conclusions justified.
The study is reported in sufficient detail.
The results are comprehensively presented.
The attached figures are fully sufficient. Comprehensive materials are included in the supplement.
I really appreciate it.
The discussion has been divided into sections, which makes it very easy to read. Generally, the discussion was well written and is very interesting to read.
The paper includes an exhaustive literature, both older and recent.


My native language is not English. For me the text is very understandable and seems to be professionally written.

Thank you for the work.

Experimental design

The study is properly designed. It is written in an understandable way. Advanced statistical analyses have been performed to analyse the obtained data.

Validity of the findings

no comment

Additional comments

Headline of figure 2. I would ask the authors to correct a typo in the description of this figure (make ‘Poland’ instead of ‘Polan’).

Reviewer 2 ·

Basic reporting

The paper presents interesting research conducted on linear landscape elements in Poland. The habitat compositions of highways verges, railway embankments and levees were determined. Recommendations were set that could improve the current management of linear landscape structures. Priority actions were highlighted that could positively affect the state of ecosystem services and take care of pollinator species diversity and conservation. The manuscript identifies very interesting and important issues.
Figures are understandable and sufficient.
Included in the manuscript is all the literature to assist in the study of the issues.
The text is understandable written in correct English.

Experimental design

The format of the manuscript follows the structure of the journal.
The abstract is concise and clearly states the aim of the work, the results and recommendations of the research.
Data was collected from publicly available databases.
The analyses were carried out correctly. For each study point, the type of linear landscape elements, coordinates and administrative region were determined. Buffer distances were determined appropriately.
The research conducted was precisely described.
The results of the analyses are presented clearly and in detail.

Validity of the findings

Research work is well planned and executed. The collected data analyzed using appropriate statistical methods. Correctly created buffers considering the distances traveled by pollinators. Conclusions and recommendations are well formulated.

Additional comments

Please correct figure title: Figure 2 Maps of linear landscape structures in Polan replace with correct title Maps of linear landscape structures in Poland.

·

Basic reporting

The paper by Marjańska & Moroń details a well-written, designed and executed study of the landscape composition and diversity of land cover types surrounding linear features that are of interest in pollinator conservation. Clear professional English is used alongside a breadth of high quality references to the wider literature, a logical structure and very nice figures.

Experimental design

Overall I support the papers publication, but I do feel the study findings, recommendations and discussion would be strengthened by a few main suggestions below. These are primarily related to better justifying selection of buffer sizes, including some basic statistical analysis to aid interpretation of noisy results and identifying how caveats could be addressed in future studies. These main suggestions are followed by come more minor suggestions I hope the authors will find useful in making this interesting and worthwhile study of wider relevance and value.

Main suggestions:
1) Given the explanation given in L146-148 on the relationship between body size and foraging distance, did you consider a smaller buffer, e.g. 125 m? I think this needs some explanation and/or additional support from the literature (e.g. see https://doi.org/10.1038/s41598-021-87650-9, where measured foraging distances ranged 150m to > 1km. However, it is also known that foraging distance is influenced by various factors (e.g. weather, resource availability, fitness) but pollinators are often not monitored outside of the growing season so ranges could be reduced. In addition, nesting pollinators will travel relatively short distances (e.g. https://doi.org/10.1371/journal.pone.0025971)

Validity of the findings

2) Data analysis (L158-169) is largely missing. Some statistical analysis would be of value, to test whether differences between LLS type, buffer size, land cover type are significant. This does not have to be complex and could be simple One-way ANOVA or pairwise T-tests to supplement the results.

3) Related to point 1 & 2 above, Figure 6 (L205-207) does not show much difference between LLS types (overlapping boxes), and the significance needs to be tested rather than interpreted by the reader. You will likely also find more significant differences within smaller buffers (that is probably why most of the study figures are results for 250m buffers). These suggestions would improve some of the discussion points (e.g. L271-273) and recommendations made as it

4) Discussion (e.g. L323-333 and the lack of acknowledgement of study caveats: habitat fragmentation and landscape connectivity between favourable land cover types is not addressed in the present study. I’m not suggesting you do this here, but some indication of study caveats and how these could be assessed in the future (e.g. using Fragstats metrics) would be useful.

Additional comments

Minor suggestions:
The paper is well-written but there are a few instances where the text could be more concise, some examples include L17: ‘…driver of global declines in insect pollinators and the ecosystem services they provide.’ L19: remove ‘already-‘; L22; ‘as a first’; L23: ‘LLS habitat management to enhance’, L27: ‘prioritising the following’; L38: ‘structures provide opportunities for’; L39: ‘prioritised for pollinator conservation in’; L40 & 41 could be merged to ‘Highway verges could optimise pollination in agricultural and urban areas’; (many of these suggestions can be applied to the introduction also); L92: ‘…for a broad..’
Abstract: Missing a final implication related to how the recommendations could be implemented and monitored at scale using the approach you use in this study.
Highlight 5: in Poland?
L71: Not clear what is meant here, do you mean ‘different flowering plants which attract diverse pollinator communities’?
Figure 1A could be slightly more informative (I would hope most readers know where Poland is) – maybe this could be supplemented with the land cover map for Poland?
Methods text L112-127 refers to both Fig 1 & 2: perhaps these could be merged as one figure? That would also address the above comment.
L133-L134: The numbers in brackets need a bit more detail, e.g. (range: X-Y; total: Z)
L144: ‘Picked-up’ buffers needs some explanation
Figure 3 – symbol colours need to be defined.
L211: suggest ‘..and to promote ecosystem service provision’
L213: ‘has presented’ – I feel this needs to be strengthened, by following the above suggestion to supplement with statistical analyses of these relationships. You can then claim to have ‘analysed’
L250: You need a reference here, and as suggested above the rationale for buffer size selection needs to be better justified in the methods.

Reviewer 4 ·

Basic reporting

In terms of the structure of the article, it is essential to revisit the heading corresponding to the discussion with a view to enhancing its content. Details are provided in the general comments section. Furthermore, a conclusion section is required in order to summarise the main research findings, highlight the limitations of the current study, set out recommendations for practice or policy and make recommendations for further research work.

Experimental design

'no comment'

Validity of the findings

It appears that the conclusions are dispersed throughout the discussion section, which has the unfortunate effect of detracting from the clarity of the exposition and hindering a more comprehensive understanding through a final reflection on the research presented. It is therefore recommended that a dedicated section be included, which addresses the issue raised in the research and provides a rationale for the findings, including an analysis of the limitations encountered.

Additional comments

The article presents a timely research topic and offers an interesting perspective on the impact of roadside verges as environmental opportunity spaces along linear transport infrastructures. Considering these spaces as an opportunity for pollinator biodiversity is innovative and applicable to other countries. The fact that the study was conducted at the national level in Poland provides a comprehensive and valuable perspective that could serve as a basis for designing studies at the local level, which could lead to proposals for integration into infrastructure planning and design.

In general, the document is well structured and written, allowing the research process to be clearly outlined. While the overall feedback is very positive, the following points require revision:

A. Throughout the text, pollinators are referred to as a collective group without distinguishing between insects, reptiles, birds or other animals. However, in section 4.2 (lines 244 onwards) specific information is introduced on the distances that bees, flies or butterflies can fly. For the scope of this research, this level of detail may not be relevant in the way it is presented. While it is interesting to discuss the different ranges or action radii of pollinators, distinguishing between them does not add significant value to the main focus of the article. In fact, "insect pollinators include bees and most families of aculeate wasps; ants; many families of flies; many lepidopterans (both butterflies and moths); and many families of beetles. Vertebrates, mainly bats and birds, but also some non-bat mammals (rodents) and some lizards".
B. Section 4.5 "Conservation Perspectives" addresses an important issue, namely the contribution of this research to the estimation of the LLS edge area. However, a piece of information about the UK (lines 302-303) is introduced without any clear connection to the rest of the text. Unless there is a misunderstanding, such disjointed data does not contribute to the discussion or to a better understanding of the research.
C. (Lines 334-342) A discussion of traffic intensity by infrastructure type and its impact on landscape disruption is included. While the approach is interesting, the Polish transport network, like that of other countries, is organised hierarchically, with traffic flow measurements distinguishing traffic densities based on the level of urbanisation and the urban or productive centres they connect. In this sense, the claim that all motorways have 200,000 vehicles per day (in general) seems to be an oversimplification. It would be appropriate to include in the discussion a reflection on which linear spaces can be more easily integrated and which would require extraordinary measures. However, to go further in this direction, it would be necessary to have an approach where data on traffic intensity is compared, distinguishing between heavy and light vehicle flows, so that the discussion is based on reliable and verifiable data. These aspects are probably beyond the scope of this article, so the issue needs to be re-examined and better linked to the research presented.
D. In the last part of the discussion section (lines 343-354), several recommendations are listed, which would be desirable to be better formulated based on the results obtained.
It is essential to include a conclusion section to provide a clear, synthetic overview of the final scope of the research.

Reviewer 5 ·

Basic reporting

The manuscript investigates the potential role of linear landscape structures (LLSs), including highway verges, railway embankments, and levees, as features that might support pollinator populations and enhance pollination services in Poland. A notable strength of this study lies in its potential to inform conservation efforts by identifying LLSs with high suitability for pollinators. This targeted approach could help optimize resource allocation, ensuring that conservation initiatives are both efficient and impactful.

Experimental design

The experimental design primarily focuses on analyzing landscape composition around linear landscape structures (LLSs) using remotely-sensed data and land cover classifications. Buffers were applied to these structures to characterize their surrounding landscapes, and potential relationships with pollinator support were inferred based on this data.
However, the design has notable shortcomings:
- Lack of response variables: the study does not include direct measurements of pollinator abundance, diversity, or behavior, which are critical to validating claims about pollinator support.
- Correlative nature: without empirical data on pollinators, the design is limited to examining landscape composition as a proxy for habitat suitability, leaving conclusions speculative.
- Coarse resolution of data: the reliance on broad land cover categories, such as grassland or tree cover, fails to capture fine-scale habitat details essential for understanding pollinator resource availability and preferences.
- No ground-truthing: the absence of field validation for the remotely-sensed data limits the ability to verify its ecological accuracy.
To improve the experimental design, the study should incorporate direct measurements of pollinators in the targeted LLSs, compare pollinator communities in LLSs to other landscape features, and use more detailed habitat classifications to better link landscape composition to ecological outcomes.

Validity of the findings

The validity of the findings is limited by the lack of direct empirical data linking linear landscape structures (LLSs), such as highway verges, to pollinator populations or their ecological functions. While the study provides useful insights into the composition and potential of these landscapes, its conclusions about their suitability for pollinators are largely speculative. The reliance on remotely-sensed land cover data and broad habitat classifications further undermines the precision and ecological relevance of the findings. Without incorporating direct observations of pollinator presence, diversity, or behavior, the study's conclusions should be interpreted with caution and regarded as preliminary hypotheses rather than definitive evidence.

Additional comments

The manuscript speculates on the potential benefits of linear landscape structures (LLSs) for pollinators without presenting direct evidence. While these hypotheses are valuable, they should be framed more clearly as preliminary findings requiring further validation.
Including field-based assessments of pollinator abundance, diversity, or behavior in LLSs would greatly enhance the robustness and ecological relevance of the findings. Without these, the study risks overgeneralization.

Reviewer 6 ·

Basic reporting

The manuscript by Marjanska and Moron describes the composition of the landscape surrounding three types of linear landscape elements across Poland, and discusses their role as alternative habitats for insect pollinators in relation to the structure of the surrounding landscape. Despite the important conservation topic, the study has several serious shortcomings. Neither the research questions of the study, nor the rationale behind them are presented with sufficient clarity. Necessary statistical analyses are missing, as well as the respective results. The existing ones are overinterpreted.
The comments below address these issues in more detail.

Experimental design

The manuscript by Marjanska and Moron describes the composition of the landscape surrounding three types of linear landscape elements across Poland, and discusses their role as alternative habitats for insect pollinators in relation to the structure of the surrounding landscape. Despite the important conservation topic, the study has several serious shortcomings. Neither the research questions of the study, nor the rationale behind them are presented with sufficient clarity. Necessary statistical analyses are missing, as well as the respective results. The existing ones are overinterpreted.
The comments below address these issues in more detail.

Validity of the findings

The manuscript by Marjanska and Moron describes the composition of the landscape surrounding three types of linear landscape elements across Poland, and discusses their role as alternative habitats for insect pollinators in relation to the structure of the surrounding landscape. Despite the important conservation topic, the study has several serious shortcomings. Neither the research questions of the study, nor the rationale behind them are presented with sufficient clarity. Necessary statistical analyses are missing, as well as the respective results. The existing ones are overinterpreted.
The comments below address these issues in more detail.

Additional comments

The comments below address these issues in more detail.
Abstract
„We explored the potential of the surrounding landscape composition of different LLSs in Poland at spatial scales reflecting distances covered by different pollinator species as first step towards assessing the potential of habitat management on LLS verges to enhance pollinator populations and pollination services.” – The meaning of this sentence is unclear, and the sentence is too long. (And Introduction does not provide clarification either; see below.) What do you mean by ‘the potential of the surrounding landscape composition’? Your study describes the landscape composition and diversity around the LLS. How is it linked to ‘the potential of habitat management’; first step of what, and in what sense?
Highlights
How are your highlights related to the findings of your study? Highlight #1: “Linear landscape structures give an opportunity for pollinator conservation.” Your study did not address the suitability of linear landscape elements for pollinator conservation; besides, this is already known. Highlight #2: “Levee verges should be prioritized to conserving pollinators in wetlands.” How does your study show that? You are prioritizing landscape elements in wetlands, but on what basis? You have not compared the pollinator diversity etc. in levees and other landscape elements, linear or other, present in wetlands. Besides, levees occur mainly in wetlands, so we probably do not want to prioritize them in other landscapes anyway. Highlight #3: “Highway verges should be prioritized to support pollinator services for crops.” Again, it is unclear: why? What crops? Or are you referring to cultivated land in general? What results of your study support this? See also comments below. Highlight #5: “Pollinator conservation programs can be spatially scaled.” What do you mean by it, and how does your study show that? Rewrite the highlights.
Introduction
General comment: I suggest adding some general context, specifically expanding the sections related to your rationale behind your study questions. Please phrase your study questions explicitly, and structure the Results section accordingly. Some more specific comments below:
„… to our knowledge, there has been no formal quantitative assessment of landscape that surrounds the different LLSs at the level of European Union (EU) countries (for roads see: Phillips et al., 2021). Such a comparison between different LLSs is important in identifying which of the LLSs should be prioritized for nature-based landscape management. It can also provide practical insights into how LLSs can be enhanced for pollination services.“ – The gap you are addressing needs explanation. Why, and how, does the quantification and comparison of the surrounding landscape help to prioritize pollinator conservation actions on road verges and other linear landscape elements?
„LLSs are common features of the landscape, sharing construction, i.e., embankments but essentially differing in human-induced disturbances, i.e., traffic and thus its potential to support biodiversity-related ecosystem services. Therefore, we carried out a comparison of the surrounding landscape of the LLS across all of Poland,“ – Firstly, the sentence is rather unclear. LLSs share construction? I assume you mean that the ecological conditions in these habitats differ due to different ways humans use them, and as a result, the pollination service they can provide differs, too. Secondly, the logical link ’therefore’ remains obscure: you are saying that because HABITATS differ, you compare LANDSCAPE structure around these habitats. Why? No doubt the surrounding landscape affects insects living in a habitat, as do the conditions in the habitat, but how are these effects related, in your opinion, with each other and with the pollination service of the ecosystem?
Materials and Methods and Results
The landscape data is correctly collected and the landscape elements and buffers chosen according to best knowledge of the ecological requirements of the main pollinator groups. However, there are several issues with the data analyses.
“2.4. Data analysis and visualization” – Neither here nor in the Results or Figures, can I see the description of the statistical analyses needed to compare the landscape composition around different LLSs, or the respective statistics. The analyses to compare the landscape diversity indices among different LLSs are also missing. It seems that the current results are based purely on visual inspection of the Figures. Therefore, we cannot be sure what the results actually are. These analyses need to be performed and the results rewritten accordingly. Some other comments below.
„3.1. Surrounding landscape of LLSs vs. Poland’s landscape“ – The titles of the paragraphs should reflect your research questions clearly. What research question does this answer to?
„3.3. Surrounding landscape of the LLS according to cover distribution shapes“. – The term ’cover distribution shape’ is not explained previously, nor is it explained how is it related to your study questions or pollinator conservation.
„The distributions of tree, grass, crop and built cover are highly right skewed for all LLS 250 m buffers around the sampling points (Fig. 5).“ – Not clear what it means and why it matters. Please also explain on Fig. 5: what is on x-axis and what is on y-axis.
„3.4 Surrounding landscape diversity of LLSs
The inverse Simpson index revealed that the landscape diversity of seven land covers (Fig. 3) within a 250 m buffer around sampling points was the highest for highways (D = 2.44; 207 Fig. 6), lower for levees (D = 2.23; Fig. 6) and lowest for railways (D = 2.03; Fig. 6).“ – I cannot see any statistical analyses to support this claim. Based on Figure 6, the diversity of the surrounding landscape does not seem to differ significantly among the studied types of LLSs. Why only 250m buffer, what about other buffer radii? Once more, how is the landscape diversity related to the possible pollinator conservation activities on LLSs?
Discussion
I am not reviewing Discussion thoroughly, as the data need proper analyses, and Discussion needs rewriting afterwards. Just a few points.
I suggest rewriting and refocusing your whole discussion in a more general context, specifically focusing on the limits of the conclusions, and implications, that can be made based on your study. It can also be shortened and made more concise.
You are suggesting that, in order to provide crop pollination, highway verges should be specifically managed for pollinators, because „highway verges may provide pollination services to the largest proportion of crop land cover”. How does the fact that highways are surrounded by a slightly larger proportion of fields than railways or levees make highway verges a (potentially) good source of pollination service? Highway verges are not good habitats for pollinators (as small country roads can sometimes be). Usually, road verges in agricultural landscapes are quite narrow, and heavily disturbed from both edges, if neighbored by intensively managed agricultural fields. Roadsides are associated with higher bee mortality due to traffic, they also have higher levels of pollution from fuel and pesticides. Road sides usually need to be frequently mown due to safety regulations, making them flower-poor. With modern traffic loads on highways, only a few pollinator taxa can live under such circumstances. Furthermore, considering the relatively small area of road verges compared to the rest of the landscape, field sizes, and need for pollination, it is highly improbable that the relatively small number of pollinators that can potentially inhabit highway verges would be sufficient to considerably improve the necessary pollination service, no matter how pollinator-friendly we manage these verges. Your suggestion, therefore, is supported neither by previous knowledge about crop yields in agricultural landscapes, nor by your study. Heterogeneous landscapes containing larger areas of suitable habitat, with more diverse conditions, and little human impact are needed to considerably increase crop yields in pollinator deficient crops. In general, I suggest rewording and refocusing your whole discussion in a broader context, specifically focusing on the limits of the conclusions and implications that can be made based on your study.

---

## Round 0.2 · accepted · Accept

· Academic Editor

Accept

Dear Dr. Moroń, I congratulate you on the acceptance of your article for publication. I hope that the practical results of your research will be applied to the protection of insects in many countries around the world. I wish you success in your further research.

Reviewer 4 ·

Basic reporting

No comment.

Experimental design

No comment.

Validity of the findings

No comment.

Additional comments

In my opinion, the authors have addressed the issues that were identified for improvement in the review. I would like to express my gratitude to the authors for their response to each of the comments made. It is my conviction that the article has undergone a substantial enhancement in its overall quality.
The article is generally well written, but as English is not my mother tongue, there may be nuances that need to be reviewed by a specialist for publication.